# Conditional generation of molecules from disentangled representations

## Abstract

Though machine learning approaches have shown great success in estimating properties of small molecules, the inverse problem of generating molecules with desired properties remains challenging. This difficulty is in part because the set of molecules which have a given property is structurally very diverse. Treating this inverse problem as a conditional distribution estimation task, we draw upon work in learning disentangled representations to learn a conditional distribution over molecules given a desired property, where the molecular structure is encoded in a continuous latent random variable. By including property information as an input factor independent from the structure representation, one can perform conditional molecule generation via a "style transfer" process, in which we explicitly set the property to a desired value at generation time. In contrast to existing approaches, we disentangle the latent factors from the property factors using a regularization term which constrains the generated molecules to have the property provided to the generation network, no matter how the latent factor changes.

## 1 Introduction

Conditional molecule generation is far from being solved. The main challenge is the enormous and discrete nature of the molecules space and the fact that molecule properties are highly sensitive to molecular structure (Kirkpatrick & Ellis, 2004). Approaches to conditional generation are typically two-step, either using a model or genetic algorithm to generate candidates which are later filtered, or learning a continuous embedding of the discrete molecules and optimizing in a real-valued representation space. The former is computationally expensive, the latter performs conditional generation only very obliquely.

We propose a conditional generative model that produces candidate molecules which targeting a desired property in a single step. This approach builds on work in structured deep generative models (Kingma et al., 2014; Siddharth et al., 2017), which aim to learn a disentangled representation that factors into observed properties we want to control for, and latent factors that account for the remaining features which are either hard to annotate or irrelevant to the properties we wish to optimize.

We derive a regularizer for supervised variational autoencoders which exploits property information that we provide as supervision, ensuring that produced molecules adhere to target properties they are conditioned on. We demonstrate the ability of our model to perform accurate conditional molecule generation and a sort of "style transfer" on molecules, where a latent representation for a single molecule can have its target properties perturbed independently of its learnt structural characteristics, allowing direct and efficient generation of candidates for local optimization of molecules.

## 2 Background

Molecule discovery tasks come in two flavors. Global optimization seeks to find molecules that have a particular target property. Local optimization starts from some initial molecule and searches for molecules which have a desired property while not straying too far from the prototype. There is some overlap in methods used in the two approaches.

### 2.1 DEEP GENERATIVE MODELS FOR MOLECULES

Virtual screening methods start from a large database of possible molecules and retain the promising ones (Eckert & Bajorath, 2007), as measured by some quality function $f(\cdot)$. Machine learning approaches expand on this by dynamically generating additional candidate molecules; Segler et al. (2017) uses a stacked LSTM to produce large numbers of novel molecules which have similar characteristics to an existing database.

For properties which are expensive to evaluate, generating large sets of candidate molecules is not particularly useful. More sample-efficient global search can be achieved using Bayesian optimization methods, which use a generative model with a latent space that functions as a continuous representation of molecules (Gómez-Bombarelli et al., 2016; Kusner et al., 2017). Optimization is then carried out over this continuous representation space to find candidates which are expected to have the desired property. Local gradient-based search can also be applied on continuous latent spaces to optimize the latent representation with respect to a target property (Jin et al., 2018; Liu et al., 2018).

A challenge for these latent variable models is to reliably produce valid molecules. Character variational autoencoders (CVAEs) (Gómez-Bombarelli et al., 2016) generate molecules one character at a time, and are prone to syntactic and semantic errors; the grammar-based variational autoencoder (GVAE) (Kusner et al., 2017) and syntax-directed variational autoencoder (SD-VAE) (Dai et al., 2018) instead operate in the space of context-free and attribute grammars, respectively, to ensure syntactic validity. Other work generative models that operates on graph representations (Simonovsky & Komodakis, 2018; De Cao & Kipf, 2018; Jin et al., 2018; You et al., 2018; Liu et al., 2018), largely improving the ability to generate valid molecules.

Suppose we are given a training set of pairs $\mathcal{D} = \{(\mathbf{x}_i, \mathbf{y}_i)\}, i = 1, \ldots, N$, where $\mathbf{x}$ corresponds to molecules and $\mathbf{y}$ represents a value of some properties of the molecule $\mathbf{x}$. Assume the molecules represent an i.i.d. sample from some unknown distribution $\tilde{p}(\mathbf{x})$, which assigns high probability to molecules believed to be useful for a given task. Aside from Segler et al. (2017), which has no latent space and thus directly trains via maximum likelihood, these latent variable models are trained by optimizing a standard ELBO objective for variational autoencoders (**?**). This entails learning a stochastic encoder $q_\phi(\mathbf{z}|\mathbf{x})$ which maps molecules into a latent space, and a stochastic decoder $p_\theta(\mathbf{x}|\mathbf{z})$ for reconstructing molecules, by maximizing

$$\mathcal{L}(\theta, \phi) = \sum_{i=1}^{N} \left\{ \mathbb{E}_{q_\phi(\mathbf{z}_i|\mathbf{x}_i)}[\log p_\theta(\mathbf{x}_i|\mathbf{z}_i)] - D_{KL}(q_\phi(\mathbf{z}_i|\mathbf{x}_i)||p(\mathbf{z}_i)) \right\}. \tag{1}$$

Notably, the objective is not a function of $\mathbf{y}$: most existing generative models with latent variables do not perform direct conditional generation, and approaches for targeted molecule discovery are bolted on to the learnt model. Some, e.g. Kusner et al. (2017), are trained in an "unsupervised" manner, agnostic to any property which later may need to be optimized. Others, e.g. Gómez-Bombarelli et al. (2016); Liu et al. (2018), train the autoencoder jointly alongside a function to predict $\mathbf{y}$ from $\mathbf{z}$, hoping to guide the latent space to be also good for predicting the desired property. A recent exception is Assouel et al. (2018), which learns a deterministic autoencoder where the decoder takes the latent code and the desired property as input, using a mutual information term in training to steer the model towards generating molecules whose target properties match the input. Guimaraes et al. (2017); De Cao & Kipf (2018); You et al. (2018) instead learn generation models optimized towards specific metrics, such as drug-likeliness and solubility; the major downside is that these models must be retrained each time for a new property. In contrast, the autoencoder-based methods can be re-used to optimize towards any particular value of the property.

### 2.2 STYLE TRANSFER WITH SUPERVISED VAES

While the latent representations learned through standard VAE models perform well on the task of molecule reconstruction they do not necessarily provide interpretable factorised representations. A disentangled representation gives us additional control on the molecule generation process, allowing us to modify a single property leaving the remaining unaffected (Bengio et al., 2013a). In many cases important variation in the data is easy to annotate. For example in the case of molecule datasets we have access to different functional descriptors of the molecules obtained by chemoinformatics software such as RDKit (Landrum). Particularly useful to us here are supervised methods for learning disentangled representations (Kingma et al., 2014; Siddharth et al., 2017). These are distinct from

unsupervised disentangling approaches such as InfoGAN (Chen et al., 2016) or $\beta$-VAE (Higgins et al., 2017), which encourages the latent factor to learn a disentangled representation by modifying the objective to promote component independence.

We will learn representations that specifically disentangle molecular properties of interest which we may later want to modify. Kingma et al. (2014) demonstrates how disentangling can be used to take two MNIST images of different digits, written in different styles, and independently change the digit while holding the style constant. An analogous operation on molecules would involve holding the physical structure of a molecule (its "style") relatively fixed while modifying a salient property. Unlike (say) the style transfer example for the MNIST digits, the conditional distribution of molecules with a particular value of properties might be very diverse; for example, the QED score attempts to measure the drug-likeness of a molecule, and the set of molecules generated at high values of this score would hopefully have high probability on a large, varied set of molecules. An essential challenge here is that the property only provides a very weak signal as to the overall structure of the molecule. To account for this diversity, we model the conditional distribution with a latent variable $\mathbf{z}$, such that $p_\theta(\mathbf{x}|\mathbf{y}) = \int p_\theta(\mathbf{x}|\mathbf{z},\mathbf{y})p(\mathbf{z})d\mathbf{z}$.

Disentangling the latent code $\mathbf{z}$ from the property $\mathbf{y}$ enables style transfer. This is done by taking an initial $\mathbf{x}$, computing the posterior over the latent variable $\mathbf{z}$, and then generating a new $\mathbf{x}'$ with the property modified to have a target value $\mathbf{y}'$, with $p_\theta(\mathbf{x}'|\mathbf{y}',\mathbf{x}) = \int p_\theta(\mathbf{x}'|\mathbf{y}',\mathbf{z})p_\theta(\mathbf{z}|\mathbf{x})d\mathbf{z}$.

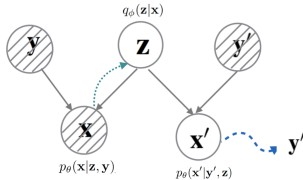

Concretely, this involves fitting a joint generative model of the form $p_\theta(\mathbf{x},\mathbf{y},\mathbf{z}) = p_\theta(\mathbf{x}|\mathbf{y},\mathbf{z})p(\mathbf{y})p(\mathbf{z})$, in which $\mathbf{y}$ and $\mathbf{z}$ are independent under the prior, and we assume a unit multivariate normal prior $p(\mathbf{z})$. To infer the latent variable $\mathbf{z}$ we will use a variational distribution $q_\phi(\mathbf{z}|\mathbf{x})$, which takes the form of a multivariate normal distribution with parameters a nonlinear function of $\mathbf{x}$, to approximate the true posterior $p_\theta(\mathbf{z}|\mathbf{x},\mathbf{y})$. This objective function

Figure 1: A demonstration of style transfer

$$\mathcal{L}_{ELBO}(\theta,\phi) = \sum_{i=1}^{N} \left\{ \mathbb{E}_{q_\phi(\mathbf{z}_i|\mathbf{x}_i)}[\log p_\theta(\mathbf{x}_i|\mathbf{y}_i,\mathbf{z}_i)] - D_{KL}(q_\phi(\mathbf{z}_i|\mathbf{x}_i)||p(\mathbf{z}_i)) \right\} \qquad (2)$$

corresponds to learning a supervised VAE (Kingma et al., 2014), and represents a fairly naïve approach to modeling a conditional distribution.

## 3 CONDITIONAL GENERATION BY DISENTANGLING

Maximizing this conditional ELBO in Eq (2) will likely yield good reconstructions of molecules from an embedding $\mathbf{z}$ (alongside the true property $\mathbf{y}$), but for properties which only weakly inform the generative model there is nothing to enforce that the variable $\mathbf{y}$ actually directly has an effect on the generative process. Since the value $\mathbf{y}$ is something we know is a derived property of the molecule $\mathbf{x}$, it is completely possible for all information about $\mathbf{y}$ to also be encoded in the representation $\mathbf{z}$, in which case there is no guarantee that the learnt likelihood $p_\theta(\mathbf{x}|\mathbf{y},\mathbf{z})$ actually takes into account the value of $\mathbf{y}$ — in fact, we know it is possible to fit variational autoencoders where the decoder simply has the form $p_\theta(\mathbf{x}|\mathbf{z})$ — and we are relying on the utility of $\mathbf{y}$ in reconstructions to see any sort of disentangling effect.

### 3.1 CONSTRAINED ELBO

In the case of conditional generation of molecules, we often have access to some oracle function $f$ (possibly non-differentiable) which for any given $\mathbf{x}$ outputs a property estimate $\mathbf{y}$, for instance, the chemoinformatics software RDKit (Landrum). Since for conditional generation our ultimate goal is to generate a molecule $\mathbf{x}$ for any given target property $\mathbf{y}_0$, which then actually has $f(\mathbf{x}) = \mathbf{y}_0$, we can reframe the problem by introducing hard constraints on the generated values, i.e. if restricting to values of $\mathbf{y}$ in the training set,

$$\max_{\theta,\phi} \mathcal{L}_{ELBO}(\theta,\phi)$$

$$\text{subject to } \mathbb{E}_{\mathbf{x}\sim p(\mathbf{x}|\mathbf{y}_i)}[\mathbb{I}[f(\mathbf{x}) = \mathbf{y}_i]] = 1$$

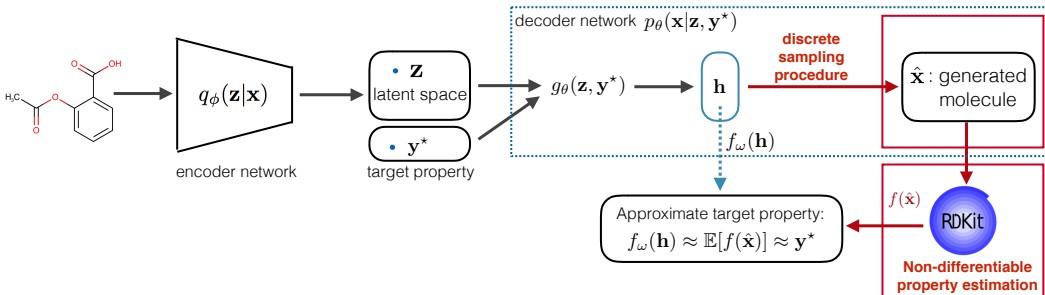

Figure 2: Setting and modeling pipeline for conditional generation of molecules, with supervision provided via an external property prediction oracle. Red lines correspond to non-differentiable components, including both a potentially complex sampling process and the property prediction itself. The blue dashed line corresponds to the approximate property predictor, which aims to predict the expected value of the property from a continuous relaxation, marginalized over the sampling process.

for all $i = 1, \ldots, N$. This is an unreasonably hard constraint, unlikely to be satisfied by any distribution other than one which simply places a point mass on the single training $\mathbf{x}_i$ associated with $\mathbf{y}_i$, but we can relax it by considering that (unlike the molecular space $\mathbf{x}$) the property space $\mathbf{y}$ is typically smooth, as many properties are continuous-valued and correspond to a human-interpretable scale. Following Ma et al. (2018) and Hu et al. (2017), we reframe the constraint as a soft penalty on the ELBO,

$$\mathcal{L}(\theta, \phi) = \mathcal{L}_{ELBO}(\theta, \phi) - \frac{\lambda_1}{2} \sum_{i=1}^{N} \mathbb{E}_{\hat{\mathbf{x}} \sim p_\theta(\mathbf{x}|\mathbf{y}_i)} \|f(\hat{\mathbf{x}}) - \mathbf{y}_i\|^2 \tag{3}$$

so that they are consistent with the property prediction, i.e., as we have an oracle function $f$ which enable us to access the property of any generated data, we can explicitly add a soft constraint to our loss function to provide explicit guidance for the generative model such that $f(\hat{\mathbf{x}}) = \mathbf{y}$. This constraint is expected to hold for any pair $(\mathbf{x}, \mathbf{y})$ we may happen to come across, not just those in the training data. We also show optimizing the relaxed constraint is equivalent to maximizing mutual information with the target $\mathbf{y}_i$ and generated molecule $\hat{\mathbf{x}}$; for details see appendix Section 6.1.

## 3.2 APPROXIMATING THE PROPERTY PREDICTOR

Introducing the regularizer as in Eq. (3) implicitly guides the reconstruction to take into account the property information, such that the reconstructed data should exhibit properties which match the input properties it is conditioned on. However, existing implementations of $f$ are often non-differentiable or CPU-bound, and $\hat{\mathbf{x}}$ are discrete samples from a categorical distribution, all of which means the gradient of the regularizer can't flow back to the generator. This is outlined in Figure 2. To enable the gradient based methods on GPUs during training and avoid discrete sampling, one approach would be to first fit a differentiable approximation to $f$, and then use either a Gumbel-softmax relaxation (Jang et al., 2016) or tricks like a "straight-through" estimator (Bengio et al., 2013b) as a continuous approximation for the discrete samples. Instead, we propose bypassing the discrete sampling step entirely and learning a function $f_\omega$ that can map from a learned representation of the molecules directly to molecules property (Hu et al., 2017).

To do this, we take as input the last hidden layer of the decoder network which parameterizes $p_\theta(\mathbf{x}|\mathbf{z}, \mathbf{y})$, denoting this deterministic transformation as $g_\theta(\mathbf{z}, \mathbf{y})$. For the grammar VAE and the syntax-directed VAE, this last layer $\mathbf{h} = g_\theta(\mathbf{z}, \mathbf{y})$ is the output of a recurrent layer that generates logits corresponding to unmasked and unnormalized log probabilities for each character at each position in the string; see Kusner et al. (2017) and Dai et al. (2018) for details on the implementation of the somewhat complex sampling process in the decoder. Ideally, $f_\omega$ would estimate the property distribution obtained by marginalizing out the discrete sampling step, with

$$f_\omega(\mathbf{h} \equiv g_\theta(\mathbf{z}, \mathbf{y}_0)) \approx \mathbb{E}_{p_\theta(\mathbf{x}|\mathbf{z}, \mathbf{y}_0)}[f(\mathbf{x})], \tag{4}$$

where we condition on $\mathbf{z}$, and $\mathbf{y}_0$ refers to an arbitrary input target property.

Assuming the approximation in Eq. (4), we have

$$\mathbb{E}_{p_\theta(\hat{\mathbf{x}}|\mathbf{y}_i)}\|f(\hat{\mathbf{x}}) - \mathbf{y}_i\|^2 = \mathbb{E}_{p(\mathbf{z})}\left[\mathbb{E}_{p_\theta(\hat{\mathbf{x}}|\mathbf{y}_i,\mathbf{z})}\|f(\hat{\mathbf{x}}) - \mathbf{y}_i\|^2\right] \approx \mathbb{E}_{p(\mathbf{z})}\|f_\omega(g_\theta(\mathbf{z},\mathbf{y}_i)) - \mathbf{y}_i\|^2,$$

an expectation over a real-valued variable which does not depend on any of the parameters we are estimating, meaning we can use a simple path estimate of the gradient with respect to $\theta, \omega$ by exchanging the gradient with the expectation. We thus define a regularization term

$$\mathcal{L}_{disent}(\theta, \omega) = \frac{\lambda_1}{2}\sum_{i=1}^{N}\mathbb{E}_{p(\mathbf{z})}\|f_\omega(g_\theta(\mathbf{z},\mathbf{y}_i)) - \mathbf{y}_i\|^2 \qquad (5)$$

which can be used as a drop-in replacement for the non-differentiable penalty term in Eq. (3), yielding a candidate objective function

$$\mathcal{L}(\theta, \phi) \approx \hat{\mathcal{L}}_\omega(\theta, \phi) = \mathcal{L}_{ELBO}(\theta, \phi) - \mathcal{L}_{disent}(\theta, \omega) \qquad (6)$$

### 3.3 Learning the property estimator jointly with generative model

While one could imagine attempting to learn $f_\omega$ jointly with $\phi, \theta$ by direct optimization of Eq. (6), in practice this is very unstable, as values of $g_\theta(\mathbf{z}, \mathbf{y}_i)$ early in training may correspond to very poor generated molecules $\hat{\mathbf{x}}_i$ which may not have properties at all similar to $\mathbf{y}_i$. This can be sidestepped by training the property estimator jointly as part of an extended generative model on $[\mathbf{x}, \mathbf{y}]$.

We note that the property estimator $f_\omega$ parameterizes a probability distribution $p_\omega(f(\mathbf{x})|\mathbf{z}, \mathbf{y}_0)$, where $\mathbf{x} \sim p_\theta(\mathbf{x}|\mathbf{z}, \mathbf{y}_0)$ and f is the oracle function that $f(\mathbf{x}) = \mathbf{y}$. With a Gaussian distribution over the error, we can consider

$$p_\omega(f(\mathbf{x})|\mathbf{z}, \mathbf{y}_0) = \mathcal{N}(f(\mathbf{x})|f_\omega(g_\theta(\mathbf{z}, \mathbf{y}_0)), \lambda_2^{-1}\mathbf{I}) \qquad (7)$$

for small, fixed $\lambda_2$. Therefore, we propose defining a new ELBO based on a joint autoencoder for $\{f(\mathbf{x}_i), \mathbf{y}_i\}$, albeit with a factorization such that the input $\mathbf{y}_i$ bypasses the encoder and is passed directly into the decoder, with a joint likelihood

$$p_{\{\theta,\omega\}}(\mathbf{x}_i, f(\mathbf{x}_i)|\mathbf{z}_i, \mathbf{y}_i) = p_\omega(f(\mathbf{x}_i)|\mathbf{z}_i, \mathbf{y}_i)p_\theta(\mathbf{x}_i|\mathbf{z}_i, \mathbf{y}_i). \qquad (8)$$

This yields a joint ELBO for the training set of

$$\mathcal{L}_{ELBO}(\omega, \theta, \phi) = \sum_{i=1}^{N}\mathbb{E}_{q_\phi(\mathbf{z}|\mathbf{x}_i)}\left[\log\frac{p_\omega(f(\mathbf{x}_i)|\mathbf{z}, \mathbf{y}_i)p_\theta(\mathbf{x}_i|\mathbf{z}, \mathbf{y}_i)p(\mathbf{z})}{q_\phi(\mathbf{z}|\mathbf{x}_i)}\right]. \qquad (9)$$

Note that we can rewrite this ELBO as a function of the previous one, with

$$\mathcal{L}_{ELBO}(\omega, \theta, \phi) = \mathcal{L}_{ELBO}(\theta, \phi) - \frac{\lambda_2}{2}\sum_{i=1}^{N}\mathbb{E}_{q_\phi(\mathbf{z}|\mathbf{x}_i)}\|f_\omega(g_\theta(\mathbf{z}, \mathbf{y}_i)) - \mathbf{y}_i\|_2^2, \qquad (10)$$

where we also see that $\mathcal{L}_{ELBO}(\omega, \theta, \phi) \leq \mathcal{L}_{ELBO}(\theta, \phi)$, allowing us to define an objective

$$\hat{\mathcal{L}}(\omega, \theta, \phi) = \mathcal{L}_{ELBO}(\omega, \theta, \phi) - \mathcal{L}_{disent}(\theta, \omega), \qquad (11)$$

which is a lower bound on Eq. (6). Notice the two terms we have added to the original ELBO are quite similar, differing only in choice of distribution: for learning $f_\omega$, we wish to use values of $\mathbf{z}$ simulated form the approximate posterior $q_\phi(\mathbf{z}|\mathbf{x})$, whereas for enforcing a constraint across all possible generations we simulate $\mathbf{z}$ from the prior $p(\mathbf{z})$.

### 3.4 Gradient estimation

As the regularizer $\mathcal{L}_{disent}(\theta, \omega)$ encourages disentangling by constraining the molecules generated from $\mathbf{y}_i$ to have property $\mathbf{y}_i$ no matter what value $\mathbf{z}$ takes, we found that it does not necessarily evaluate at meaningful values of $\mathbf{z}$ when sampled randomly from $p(\mathbf{z})$. This roughly corresponds to the notion that not all combinations of "style" and property are physically attainable; ideally for style transfer we would like the generated molecule to stay "close" in structure to the original molecule that we intended to modify. When estimating (gradients of) the soft constraint term $\mathcal{L}_{disent}(\theta, \omega)$, we found it advantageous to use samples of $\mathbf{z}$ which correspond to encodings of actual data points, as

| | QM9 | | | | ZINC | | | |
|---|---|---|---|---|---|---|---|---|
| Model | Reconstruction % | Valid% | Unique % | Novel % | Reconstruction % | Valid% | Unique % | Novel % |
| CVAE Gómez-Bombarelli et al. (2016) | 3.61 | 10.30 | - | 90.0 | 44.6 | 0.70 | - | 100 |
| GVAE Kusner et al. (2017) | 96.00 | 60.20 | - | 80.90 | 53.70 | 7.20 | - | 100 |
| SD-VAE Dai et al. (2018) | 97.84 | **98.40** | 99.28 | 91.97 | 76.20 | **43.50** | - | - |
| Sup-VAE-1-GRU | 97.53 | 93.66 | 91.30 | 92.05 | 74.12 | 32.84 | 94.61 | 100 |
| CGD-VAE-1-GRU | **99.27** | 95.61 | 93.65 | 87.87 | **88.64** | 29.00 | **99.24** | 100 |
| Sup-VAE-3-GRU | 97.81 | 97.90 | 95.09 | 89.47 | 82.40 | 36.16 | 86.26 | 100 |
| CGD-VAE-3-GRU | 99.31 | 97,80 | **98.77** | **96.21** | 81.80 | 37.78 | 98.75 | 100 |

Table 1: Reconstruction performance and generation quality (Valid, Unique, Novel).

opposed to random samples from the prior. We approximate expectations with respect to $p(\mathbf{x})$ by looking at the so-called marginal posterior; we note that

$$p(\mathbf{z}) = \int p_\theta(\mathbf{z}|\mathbf{x})p_\theta(\mathbf{x})d\mathbf{x} \approx \frac{1}{N}\sum_j p_\theta(\mathbf{z}|\mathbf{x}_j) \approx \frac{1}{N}\sum_j q_\phi(\mathbf{z}|\mathbf{x}_j),$$

where the first approximation uses the empirical data distribution as an approximation to the model marginal $p_\theta(\mathbf{x})$, and the second uses our variational posterior approximation $q_\phi(\mathbf{z}|\mathbf{x})$. We define this quantity as $q(\mathbf{z}) = \frac{1}{N}\sum_j q_\phi(\mathbf{z}|\mathbf{x}_j)$, a mixture of Gaussians, which we can sample from by drawing random values from our dataset and then drawing from their encoding distributions.

When we use this in estimating gradients of the soft constraint, we can use samples from the same minibatch, exactly corresponding to a property transfer task. That is, for any particular $y_i$ in the dataset, we can estimate

$$\mathbb{E}_{p(\mathbf{z})}\nabla_{\theta,\omega}\|f_\omega(g_\theta(\mathbf{z},\mathbf{y}_i)) - \mathbf{y}_i\|^2 \approx \mathbb{E}_{q(\mathbf{z}_j|\mathbf{x}_j)}\nabla_{\theta,\omega}\|f_\omega(g_\theta(\mathbf{z},\mathbf{y}_i)) - \mathbf{y}_i\|^2.$$

for any uniformly randomly sampled $j \neq i$. By sampling $\mathbf{z}_j$ from $q(\mathbf{z}_j|\mathbf{x}_j)$ where $j \neq i$, we make sure that all the label information decoder is receiving comes from the actual $\mathbf{y}_i$ that is feed to the decoder and $\mathbf{z}_j$ does not include any information about label. This can be evaluated easily by simply evaluating the penalty term of Eq. (10) twice per minibatch; once as in Eq. (10), and once to approximate $\mathcal{L}_{disent}(\theta,\omega)$ by permuting the properties in the minibatch to be assigned to incorrect molecules. We detail the training algorithm in Section 6.2 of the appendix.

## 4 EXPERIMENTS

We experiment with the QM9 dataset (Ramakrishnan et al., 2014), that contains 134k molecules with up to 9 heavy atoms, and the ZINC dataset (Sterling & Irwin, 2015) containing 250k drug-like molecules. Our goal here is two-fold: we would like to understand (1) whether a supervised variational autoencoder is capable of learning suitable conditional distributions over molecules, and (2) to what extent this task is assisted by the additional regularization term corresponding to the soft constraint.

We represent molecules using the one-hot encoding of their SMILES production rules (Kusner et al., 2017) and add a semantic constraint (Dai et al., 2018) on the decoder network to avoid generating syntactically correct but semantically invalid molecules. We use 80 production rules to describe molecules and set the maximum SMILES sequence length to 100 for the QM9 dataset and 278 for the Zinc dataset. We experiment with the logP property of the molecules (Wildman & Crippen, 1999). We use the same encoder and decoder network structure as Dai et al. (2018) with the only difference that our decoder takes as input the concatenation of $\mathbf{y}, \mathbf{z}$. We give the details of the architecture in the appendix section 6.2.

We evaluate the reconstruction accuracy and the quality of the molecules generated by our method, which we denote by CGD-VAE (conditional generation with disentangling) and compare against CVAE (Gómez-Bombarelli et al., 2016), GVAE (Kusner et al., 2017), and SD-VAE (Dai et al., 2018). We explore its conditional generation performance in two settings: controlling only the property value and controlling both the property value and the molecule structure to what can be seen as property transfer. We took the results of CVAE, GVAE from the literature. For SD-VAE we used the authors code with the default values to generate results for QM9 since these were not available for QM9. We also implemented supervised VAE versions of SD-VAE which we denote Sup-VAE-X-GRU ($X \in \{1, 3\}$, denotes GRU layers) and which can do conditional generation.

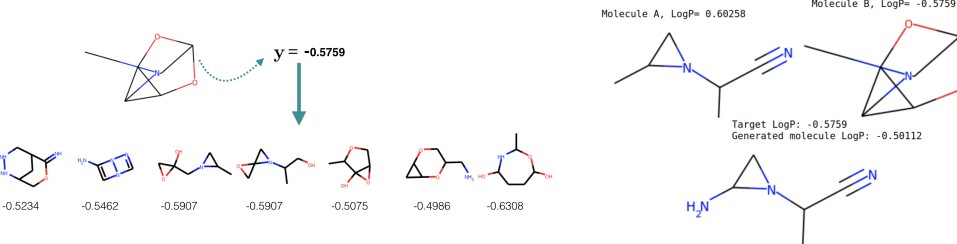

Figure 3: Conditional generation given the desired logP=-0.5759, row molecules have a logP within a 15% range of the desired one.

Figure 4: Property transfer

Before proceeding with the experiments we will give some additional details on how we do conditional generation from $p_\theta(\mathbf{x}|\mathbf{y}_0)$ given the target property $\mathbf{y}_0$. Instead of marginalizing over the prior $p(z)$, we mirror the approach taken during training and integrate over an approximation to the marginal inference distribution $q_\phi(\mathbf{z}) = \frac{1}{N}\sum_{i=1}^{N} q_\phi(\mathbf{z}|\mathbf{x}_i)$ which better characterizes where the mass of the dataset is in the latent space. However, as $N$ is large and we do not wish to keep the entire dataset available at test time, we approximate $q_\phi(\mathbf{z})$ with an isotropic Gaussian distribution $\hat{q}_\sigma(\mathbf{z}) = \mathcal{N}(\mathbf{z}|\mathbf{0}, \sigma^2\mathbf{I})$. We estimate $\sigma$ for each model by Monte Carlo samples from $q_\phi(\mathbf{z})$. For the supervised VAE without the soft constraint regularizer this yields $0.053$ for QM9 and $0.118$ for ZINC. For our model with the soft constraint we get $0.0354$ for QM9 and $0.096$ for ZINC. We do conditional generation of $\mathbf{x}$ given $\mathbf{y}_0$ by sampling from $p_\theta(\mathbf{x}|\mathbf{y}_0) = \int \hat{q}_\sigma(\mathbf{z})p_\theta(\mathbf{x}|\mathbf{z}, \mathbf{y}_0)d\mathbf{z}$.

We evaluate *reconstruction performance* in terms of the correctly reconstructed molecules on test sets of size 10k for QM9 and 5k for ZINC, for the latter we used the default test set. We evaluate the generated molecules' quality by the percentage of valid, unique (i.e. percentage of unique molecules among the generated valid molecules) and novel (i.e. percentage of molecules never seen in the training set among the generated molecules) molecules. We estimate these quantities by sampling 10k (5K for ZINC) $\mathbf{z}$ from the $\hat{q}_\sigma(\mathbf{z})$ and coupling each one of them with a logP value, $\mathbf{y}$, randomly selected from the test set, and we subsequently decode the $\mathbf{z}, \mathbf{y}$ concatenation. We can see that our model has a better reconstruction performance compared to the baselines while in some cases generating slightly less valid molecules table 1. In terms of the three quality measures achieves an excellent performance across all three metrics being always one of the two best performing methods for any metric.

To visualise how the *conditional generation* operates we randomly sample from the test set some molecule and obtain its property value $\mathbf{y}_0$. We then draw 50 random samples $\mathbf{z}_i$ from $\hat{q}_\sigma(\mathbf{z})$ and decode the $[\mathbf{z}_i, \mathbf{y}_0]$ vectors. Among the generated valid molecules we compute the percentage of those that have a property value $\mathbf{y}_i$ that is within a 15% range from the $\mathbf{y}_0$ property value. In Figure 3 we present the molecules obtained for a test molecule that had a logP of $-0.5759$. Out of the 50 generated molecules 46 were valid of which we give the five that were within a 15% range from the $\mathbf{y}_0$ value in Figure 3. As we can see we get molecules that are structurally very different from the original one yet they have similar logP value.

To quantify the quality of the conditional generations we measure the correlation between the property value we obtain by the conditional generation and the property value on which we conditioned the generation. We randomly sample 1000 $\mathbf{y}$ values from the test set and 1000 $\mathbf{z}$ values from the approximate learned prior $\hat{q}_\sigma(\mathbf{z})$. We decode each pair, obtain $\hat{\mathbf{x}} \sim p_\theta(\mathbf{x}|\mathbf{y}, \mathbf{z})$, and then measure the correlation of the original $\mathbf{y}$ with the $\hat{\mathbf{y}}$ of generated $\hat{\mathbf{x}}$. In Table 2, we give the correlation estimates for our method and the Sup-VAE baselines. As we can see our method has a considerably higher correlation score between the input and the obtained property than Sup-VAE. Conditional generation seems considerably harder for the ZINC dataset for all methods.

To visualise the *style transfer* behavior of our model we randomly sample two molecules $\mathbf{x}_A, \mathbf{x}_B$ from the test set. We then sample $\mathbf{z}_A$ from the learned posterior $q_\phi(\mathbf{z}|\mathbf{x}_A)$. We subsequently decode $[\mathbf{z}_A, \mathbf{y}_B]$, $\mathbf{y}_B$ is the property of $\mathbf{x}_B$, and get a new molecule $\hat{\mathbf{x}}_{AB}$. Ideally, the obtained molecule $\hat{\mathbf{x}}_{AB}$ should have a property value (logP) close to the target $\mathbf{y}_B$ and be similar to $\mathbf{x}_A$. In Figure 4 we give one such example. To put the results into context in Figure 8 in appendix, we give the results of a virtual screening method, where we select from the full dataset five molecules which are structurally

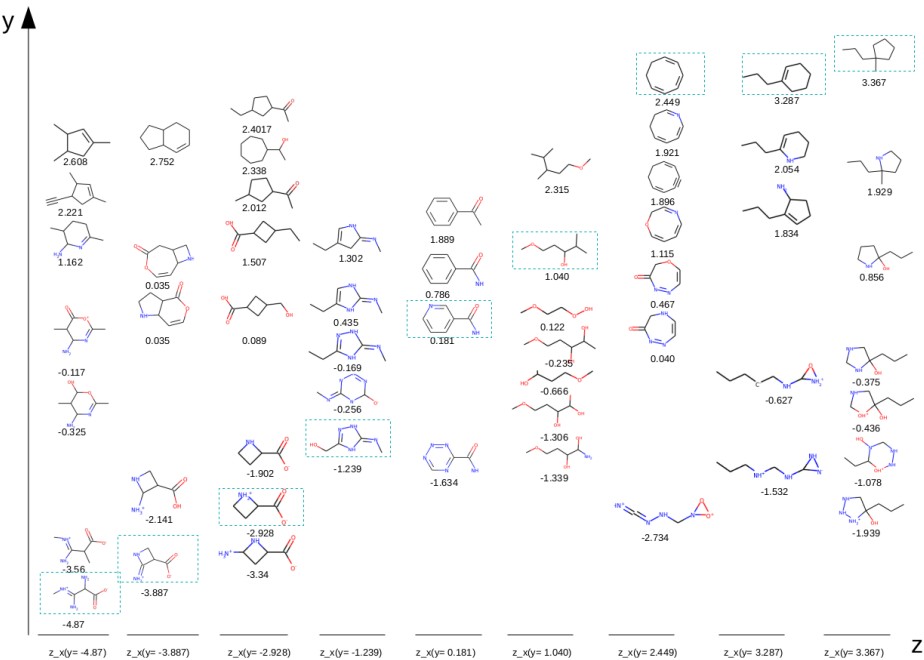

Figure 5: Style transfer. The $\mathbf{z}$ of the nine real molecules placed in the x-axis is combined with 11 $\mathbf{y}$ property values, sampled in [-4.9, 4.9], the resulting pair is decoded to a molecule.

similar to $\mathbf{x}_A$ and have logP values close to $\mathbf{y}_B$. As we can see the molecule that our model generates is a new one.

To quantify the style transfer performance we proceed in exactly the same manner as we did to quantify the conditional generation performance. However, now instead of sampling $\mathbf{z}$ from the approximate learned prior, $\hat{q}_\sigma(\mathbf{z})$, we first sample some $\mathbf{x}$ from the test set and then we sample $\mathbf{z}$ from the learned posterior $q(\mathbf{z}|\mathbf{x})$. The results are in the second column of Table 2. As we can see the correlation values are now lower than the ones we obtained in the simple conditional generation case. This can be explained by the fact that now we are forcing a specific combination of structure, $\mathbf{z}$ comes from a real molecule, and property, which might simply be physically infeasible since the molecule space is discrete and not all combinations are possible. In addition, as it was the case for the conditional generation, style transfer is considerably more difficult for the ZINC dataset.

We further explore the style transfer and visualize how our model covers the combined space of molecule structure and properties. We sample nine molecules from the QM9 test set, and get their $\mathbf{z}$ encodings. For each such encoding we decode the vectors $[\mathbf{z}, y], y \in [-4.9, 4.9]$, with the $y$ (logP) interval sampled at 11 points. We give in Figure 5 the resulting valid molecules, each column there corresponds to one of the nine original molecules, the ones surrounded by dotted rectangle, and their decodings with different logP values.For each original molecule we give the generated molecules ordered along the y axis according to the y property that they actually exhibit. The x-axis does not provide an ordering of the original molecules according to z, in fact we have ordered the original molecules by their y property. As we can see not all $(\mathbf{z}, y)$ combinations produce a result. These holes can be explained either by the physical infeasibility of the combination and/or a limitation of the learned model.

We can use conditional generation to control in a fine manner the value of the desired property, to what can be seen as direct *property optimization*. We visualise the level of control we have on an experiment with a single molecule (with logP is -1.137), which we randomly sample from the test set. We obtain its $\mathbf{z}$ encoding and perform generations with increased logP taking values in 1000 point grid in $[-1.137, 4.9]$. We then decode $[\mathbf{z}, \mathbf{y}_i]$ and compute the logP value of the generated molecules. Among the 1000 generated molecules only 19 are unique. We get an increase of logP of a very discrete nature, Figure 6. As already discussed not all combinations of structure and properties are possible. The generated molecules themselves are shown in the supplemental material.

| | Model | $\mathbf{z} \sim \hat{q}_\sigma(\mathbf{z})$ | $\mathbf{z} \sim q(\mathbf{z}\vert\mathbf{x})$ |
|---|---|---|---|
| QM9 | Sup-VAE-1-GRU | 0.5420 | 0.2526 |
| | CGD-VAE-1-GRU | 0.7185 | 0.5005 |
| | Sup-VAE-3-GRU | 0.6958 | 0.4204 |
| | CGD-VAE-3-GRU | 0.7414 | 0.4715 |
| ZINC | Sup-VAE-1-GRU | 0.2301 | 0.0481 |
| | CGD-VAE-1-GRU | 0.3877 | 0.0880 |
| | Sup-VAE-3-GRU | 0.3514 | 0.1808 |
| | CGD-VAE-3-GRU | 0.3966 | 0.1559 |

Table 2: Correlation between the desired input property and the obtained property . $\mathbf{z} \sim \hat{q}_\sigma(\mathbf{z})$ corresponding to conditional generation), and $\mathbf{x}, \mathbf{z} \sim q(\mathbf{z}\vert\mathbf{x})$ to property transfer case.

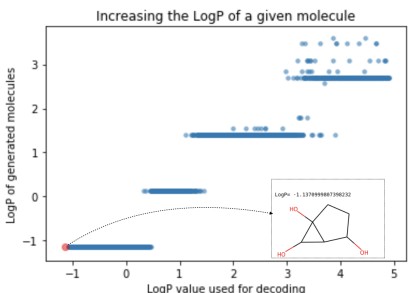

Figure 6: Property optimization. Given a start molecule (red), we combine its $\mathbf{z}$ with 1000 logP values and decode (blue)

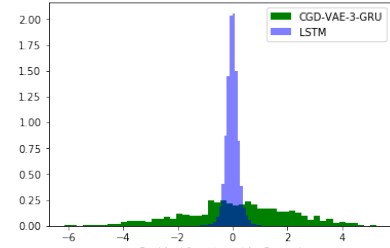
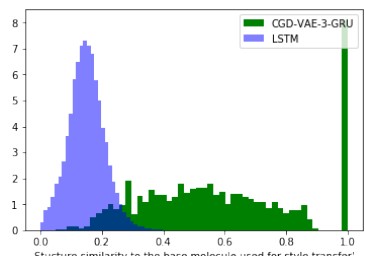

Figure 7: A comparison of simulated logP values and Tanimoto similarity to a target on the ZINC dataset. While the stacked LSTM model has high accuracy in terms of matching the desired property, it would require drawing many samples before finding any close matches to any particular desired prototype. The CGD-VAE-3-GRU model represents a middle ground between a standard VAE model which does not condition on the property, and the stacked LSTM model which does not learn a reusable representation.

## 4.1 CONDITIONAL LSTM BASELINE

Finally, we consider a variant of the stacked LSTM model of Segler et al. (2017), with no latent space, where the model is modified to take a target logP value as an additional input at each generation step. This model forms a very strong baseline for many distribution matching tasks (Liu et al., 2018; **?**), though as best we are aware this has never been used directly for conditional generation given a target property. We use a modification of the implementation provided by (**?**) with three layers and default settings, and fit the model by maximum likelihood training on $p_\theta(\mathbf{x}\vert\mathbf{y}) = \prod_{t=1}^{T} p_\theta(x_t\vert x_{1:t-1}, \mathbf{y})$.

Training on the ZINC dataset, we find the generated molecules from this model have a very high correlation 0.975 with the target logP value, greatly outperforming any of the latent variable models we consider. This suggests that such a model would be very useful for generating candidates globally, but as the model has no latent variable it is not amenable to style transfer. We observe this in Figure 7, which samples 100 candidate molecules from both the stacked LSTM model and for CGD-VAE-3-GRU, conditioning on the property of one randomly-chosen test set example, while computing the Tanimoto similarity (computed using Morgan fingerprints of radius 2) to a second randomly-chosen test set example, across 200 pairs. The VAE has higher Tanimoto similarities as it can condition on the latent variable of the target molecule, representing a trade-off against the better adherence to the target property value of the unconditioned LSTM.

## 5 CONCLUSION

We presented a single step approach for the conditional generation of molecules with desired properties. Our model allows also to condition generation on a prototype molecule with a desired high-level structure. This work thus directly inverts the traditional relationship between molecules and their properties. We found that training the deep generative models conditional on target properties, following a supervised VAE approach, does not appreciably harm the quality of the unconditional

generative model as measured by validity, novelty, and uniqueness of samples. Furthermore, we see that the additional act of regularizing the output using an approximate property predictor helps improve both reconstruction accuracy and property correlations in most combinations of tasks and datasets, particularly for the smaller QM9 dataset and for smaller models with fewer RNN layers. We also note that although none of the deep latent variable models are competitive with an LSTM baseline when purely considering generation conditioned on a target property value, the low Tanimoto similarity between randomly sampled candidates and an arbitrary style transfer target makes clear that such a model is not suitable for targeted generation of candidates which are close in structure to a particular prototype.

In future work, we want to explore how to further improve the correlation between the desired input properties to the decoder, and the properties of the generated molecules. Moreover, we want also to condition on multiple properties; while this is in principle possible in our framework, we do not explore it empirically here. Modifying a single property while constraining the remaining to be close to the original can further aggravate the infeasibility problem, as not all combinations of molecular properties may even be feasible, perhaps requiring learning a dependency structure between multiple properties.

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

## 6 APPENDIX

### 6.1 THE REGULARISER AND ITS RELATION TO THE MUTUAL INFORMATION MAXIMIZATION

The soft constrain in the loss 3, in fact , is equivalent to a simple maximizing mutual information formulation between generated molecules $\hat{\mathbf{x}}$ and the target property $\mathbf{y}$ provided to the generator. Assume the true conditional distribution is $\tilde{p}(\mathbf{y}|\mathbf{x})$:

$$
\begin{aligned}
\mathbf{I}(\mathbf{y};\hat{\mathbf{x}}) &= H(\mathbf{y}) - H(\mathbf{y}|\hat{\mathbf{x}}) \\
&= H(\mathbf{y}) + \mathbb{E}_{\hat{\mathbf{x}}\sim p_\theta(\mathbf{x}|\mathbf{y})}\mathbb{E}_{\mathbf{y}'\sim\tilde{p}(\mathbf{y}|\hat{\mathbf{x}})}[\log\tilde{p}(\mathbf{y}'|\hat{\mathbf{x}})]
\end{aligned}
\tag{12}
$$

We do not know the true distribution $\tilde{p}(\mathbf{y}|\hat{\mathbf{x}})$, however, the RDKit provides an estimation $p(\mathbf{y}|\hat{\mathbf{x}})$ of the distribution assuming a Gaussian distribution over error:

$$
p(\mathbf{y}|\hat{\mathbf{x}}) = \mathcal{N}(\mathbf{y}|f(\hat{\mathbf{x}}),\lambda_1^{-1}\mathbf{I})
\tag{13}
$$

where $f$ is the molecule property estimator, i.e., RDKit. We have:

$$
\begin{aligned}
\mathbf{I}(\mathbf{y};\hat{\mathbf{x}}) &= H(\mathbf{y}) + \mathbb{E}_{\hat{\mathbf{x}}\sim p_\theta(\mathbf{x}|\mathbf{y})}\mathbb{E}_{\mathbf{y}'\sim\tilde{p}(\mathbf{y}|\hat{\mathbf{x}})}[\log\frac{\tilde{p}(\mathbf{y}'|\hat{\mathbf{x}})}{p(\mathbf{y}'|\hat{\mathbf{x}})}p(\mathbf{y}'|\hat{\mathbf{x}})] \\
&= H(\mathbf{y}) + \mathbb{E}_{\hat{\mathbf{x}}\sim p_\theta(\mathbf{x}|\mathbf{y})}[D_{kl}(\tilde{p}(\mathbf{y}'|\hat{\mathbf{x}})||p(\mathbf{y}'|\hat{\mathbf{x}})) + \mathbb{E}_{\mathbf{y}'\sim\tilde{p}(\mathbf{y}|\hat{\mathbf{x}})}\log p(\mathbf{y}'|\hat{\mathbf{x}})] \\
&\geq H(\mathbf{y}) + \mathbb{E}_{\hat{\mathbf{x}}\sim p_\theta(\mathbf{x}|\mathbf{y})}[\mathbb{E}_{\mathbf{y}'\sim\tilde{p}(\mathbf{y}|\hat{\mathbf{x}})}\log p(\mathbf{y}'|\hat{\mathbf{x}})]
\end{aligned}
\tag{14}
$$

By following the Lemma 5.1 given in Chen et al. (2016), we have

$$
\mathbf{I}(\mathbf{y};\hat{\mathbf{x}}) \geq H(\mathbf{y}) + \mathbb{E}_{\mathbf{y}\sim p(\mathbf{y}),\hat{\mathbf{x}}\sim p_\theta(\mathbf{x}|\mathbf{y})}[\log p(\mathbf{y}|\hat{\mathbf{x}})]
\tag{15}
$$

As $H(\mathbf{y})$ is constant, minimizing $\mathbb{E}_{\hat{\mathbf{x}}\sim p_\theta(\mathbf{x}|\mathbf{y}_i)}\|f(\hat{\mathbf{x}})-\mathbf{y}_i\|^2$ is equivalent to maximizing $\mathbf{I}(\mathbf{y};\hat{\mathbf{x}})$ under the assumption that $p(\mathbf{y}|\hat{\mathbf{x}})$ is close to $\tilde{p}(\mathbf{y}|\hat{\mathbf{x}})$

### 6.2 ARCHITECTURE AND TRAINING PROCEDURE DESCRIPTION

We use the same encoder and decoder network structure as Dai et al. (2018) with the only difference that our decoder takes as input the concatenation of $\mathbf{y},\mathbf{z}$. As GRU layers become computationally expensive when the sequences length increase, we also examined the model using less layer GRU. To be precise, the decoder in Dai et al. (2018) takes the from of a dense hidden layer with ReLU activation followed by three layers GRU Chung et al. (2014). We tried two different settings of decoder: in the first setting, we feed the concatenation of $\mathbf{y},\mathbf{z}$ to dense layer then apply one layer GRU, in the second setting, to enhance the effect of $\mathbf{y}$ in the decoder, we feed $\mathbf{y}$ not only to the dense layer but also to each layer of GRU. Furthermore, we set the dimension of the latent representation to 56. For the oracle function estimator $f_w$, we use the same network architecture as the encoder (there is no parameter sharing) and we add one more fully connected layer followed by a Tanh transformation.

To speed up convergence, we initialize the $f_w$ from a pre-training, where we train $f_w$ on the well-trained (maximum 500 epochs with early stopping) supervised VAE's decoder output to predict the molecules property value. We also initialize the parameters of the encoder/decoder networks with the partially trained supervised VAE model (after 40 epochs for QM9, 100 epochs for ZINC). We do not update $\omega$ and $\phi,\theta$ simultaneously, instead we do an alternate optimization. We update $\omega$ continuously for five epochs while holding $\phi,\theta$ and do the same for updating $\phi,\theta$. We set the hyper-parameter value $\lambda_1$ to 50 and $\lambda_2$ to 1. The mini-batch size is set to 300 for QM9 and 100 for ZINC. We use ADAM optimizer with learning rate 0.0001 and pytorch lr-scheduler on the validation loss. The general training algorithm is described in below algorithm block1. In our experiment, to train $f_\omega$, we skipped the second term in step 7, which means we only train $f_\omega$ on the training data but not the newly generated molecules obtained by permuting the property. The reason for this is that, during the training, we found that it is easy for the model to learn to reconstruct but hard to conditionally generate the molecules with given properties while we have no guidance of what the molecules should look like. Further more, some combination of $\mathbf{z}$ and $\mathbf{y}$ are physically not feasible. In this case, when the conditional generation is not good enough yet during the training, we end up fitting $f_\omega$ on the miss represented molecules representations and it makes the optimization harder.

---

**Algorithm 1** Training algorithm

---

1: Initialize $p_\theta(\mathbf{x}|\mathbf{z}, \mathbf{y})$, $q_\phi(\mathbf{z}|\mathbf{x})$, $f_w$
2: **for** i=1,2, ..., N (maximum epoch number) **do**
3:      **for** j = 1,2, ..., L, sample a minibatch $D = (\mathbf{X}, \mathbf{Y}) = \{\mathbf{x}_i, \mathbf{y}_i\}_{i=1}^M$ of M samples **do**
4:          randomly permute the property set $\mathbf{Y}$ to obtain $\mathbf{Y}^*$ and define a label permuted mini batch $D^* = (\mathbf{X}, \mathbf{Y}^*) = \{\mathbf{x}_i, \mathbf{y}_i^*\}_{i=1}^M$
5:          $\theta^j = \theta^{j-1} - \gamma(-\nabla_\theta \mathcal{L}_{ELBO}(\theta, \phi, \omega) + \frac{\lambda_1}{2} \sum_{(\mathbf{x}_i, \mathbf{y}_i) \in D^*} \mathbb{E}_{q(\mathbf{z}|\mathbf{x}_i)} \nabla_\theta \|f_\omega(g_\theta(\mathbf{z}, \mathbf{y}_i)) - \mathbf{y}_i\|^2)$
6:          $\phi^j = \phi^{j-1} - \gamma(-\nabla_\phi \mathcal{L}_{ELBO}(\theta, \phi, \omega))$
7:          $w^j = w^{j-1} - \gamma(\frac{\lambda_2}{2} \sum_{(\mathbf{x}_i, \mathbf{y}_i) \in D} \mathbb{E}_{q(\mathbf{z}|\mathbf{x}_i)} \nabla_\omega \|f_\omega(g_\theta(\mathbf{z}, \mathbf{y}_i)) - \mathbf{y}_i\|^2 + \frac{\lambda_1}{2} \sum_{(\mathbf{x}_i, \mathbf{y}_i) \in D^*} \mathbb{E}_{q(\mathbf{z}|\mathbf{x}_i)} \nabla_\omega \|f_\omega(g_\theta(\mathbf{z}, \mathbf{y}_i)) - \mathbf{y}_i\|^2)$

---

## 6.3 USE A FIXED PRE-TRAINED PROPERTY PREDICTION FUNCTION

We also investigate the case where we train the property prediction function $f_\omega$ on the well trained supervised VAE output and keep it fixed during the training of the main model. We give the performance in tables 3, 4. Using a fixed $f_\omega$, in terms of reconstruction and generation performance, delivers mixed results 3. However, in terms of conditional generation 4, it does perform better than the baselines but worse compared to the case where we also update $f_\omega$ during the learning.

| Model | QM9 | | | | ZINC | | | |
|---|---|---|---|---|---|---|---|---|
| | Reconstruction % | Valid% | Unique % | Novel % | Reconstruction % | Valid% | Unique % | Novel % |
| CVAE | 3.61 | 10.30 | - | 90.00 | 44.60 | 0.70 | - | 100 |
| GVAE | 96.00 | 60.20 | - | 80.90 | 53.70 | 7.20 | - | 100 |
| SD-VAE | 97.84 | 98.40 | 99.28 | 91.97 | 76.20 | 43.50 | - | - |
| Sup-VAE-1-GRU | 97.53 | 93.66 | 91.30 | 92.05 | 74.12 | 32.84 | 95.61 | 100 |
| CGD-VAE-1-GRU | 98.96 | 95.03 | 92.77 | 89.74 | 67.46 | 17.38 | 82.44 | 100 |
| Sup-VAE-3-GRU | 97.81 | 97.9 | 95.09 | 89.47 | 82.40 | 36.16 | 86.26 | 100 |
| CGD-VAE-3-GRU | 96.9 | 93.8 | 98.29 | 89.55 | 81.80 | 37.78 | 98.70 | 100 |

Table 3: Sup-VAE-1-GRU /Sup-VAE-3-GRU: supervised version of SD-VAE model where $\mathbf{y}$ is been feed to only the first layer/all layer decoder. CGD-VAE: our model, conditional generation with disentangling. The result for CVAE and GVAE are taken from the literature, "-" refers that those measures are not reported. The rest of baseline result is obtained by rerunning (SD-VAE) and editing the original code (Sup-VAE).

| | Model | $\mathbf{z} \sim \hat{q}_\sigma(\mathbf{z})$ | $\mathbf{z} \sim q_{(}\mathbf{z}|\mathbf{x})$ |
|---|---|---|---|
| QM9 | Sup-VAE-1-GRU | 0.5420 | 0.2526 |
| | CGD-VAE-1-GRU | 0.6331 | 0.3835 |
| | Sup-VAE-3-GRU | 0.6958 | 0.4204 |
| | CGD-VAE-3-GRU | 0.6665 | 0.4507 |
| ZINC | Sup-VAE-1-GRU | 0.2301 | 0.0481 |
| | CGD-VAE-1-GRU | 0.2981 | 0.0866 |
| | Sup-VAE-3-GRU | 0.3638 | 0.1818 |
| | CGD-VAE-3-GRU | 0.3765 | 0.1310 |

Table 4: Correlation between the desired input property and the obtained property when $\mathbf{z}$ is sampled from the approximate learned prior, $\mathbf{z} \sim \hat{q}_\sigma(\mathbf{z})$, (conditional generation), and when $\mathbf{z}$ is sampled from the learned posterior given some $\mathbf{x}$, $\mathbf{z} \sim q(\mathbf{z}|\mathbf{x})$, (property transfer).

## 6.4 VIRTUAL SCREENING

The figure 8 displays the results of a virtual screening method, where we select from the full dataset five molecules which are structurally similar to $\mathbf{x}_A$ in figure 4 in section 4 and have logP values close to $\mathbf{y}_B$. As we can see the molecule that our model generates is a new one.

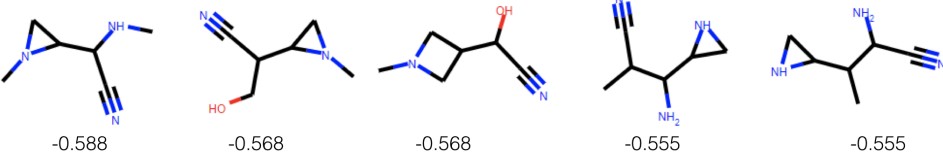

Figure 8: Molecules selected with virtual screening over the full dataset in a manner that they are structurally similar to the A molecule of figure 4 while they have a logP value which is close to the logP value of the B molecule also of figure 4

## 6.5 METRICS AS FUNCTION OF $\mathbf{y} - \mathbf{y}'$

The validity and novelty were mainly used to assess purely the generative model performance. However, it is also interesting to see if the model capable of generating valid and novel molecules if we start from an existing molecules and drift away from the original label, i.e., if we get $\mathbf{z}$ from $q(\mathbf{z}|\mathbf{x})$, and then compare different values from $p(\mathbf{x}|\mathbf{z}, \mathbf{y}')$ as $\mathbf{y}'$ moves far from $\mathbf{y}$. We randomly sample a molecule $\mathbf{x}$ whose LogP is $\mathbf{y}$ from the test set, then sample 10 $\mathbf{z}$ from $q(\mathbf{z}|\mathbf{x})$. For each such $\mathbf{z}$ we couple it with a $\mathbf{y}'$ that is different that $\mathbf{y}$, and sample 10 molecules from $p(\mathbf{x}|\mathbf{z}, \mathbf{y}')$. Eventually, for each such $\mathbf{y}'$, starting from original molecule $\mathbf{x}$, we generated 100 molecules, and we report the validity, uniqueness and novelty as a function of $\mathbf{y} - \mathbf{y}'$. The figure 9 displays result of repeating above process for 20 randomly sampled $(\mathbf{x}, \mathbf{y})$ along 100 grid points for $\mathbf{y} - \mathbf{y}'$. The result confirms that on a big data set, conditional generative models uniqueness, validity and novelty performance is not affected by the size of the modification done on the property. However, on a small dataset, uniqueness is not affected. As expected, novelty increases as the properties modification size increases and validity drops slightly as the property modification size increase.

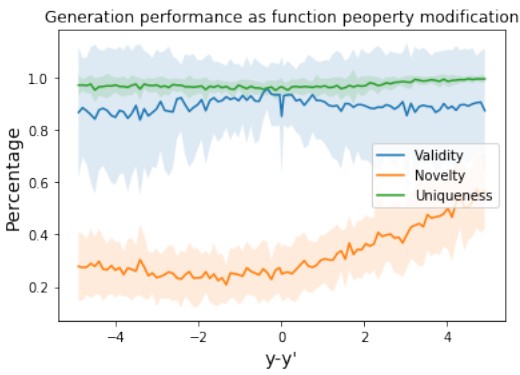

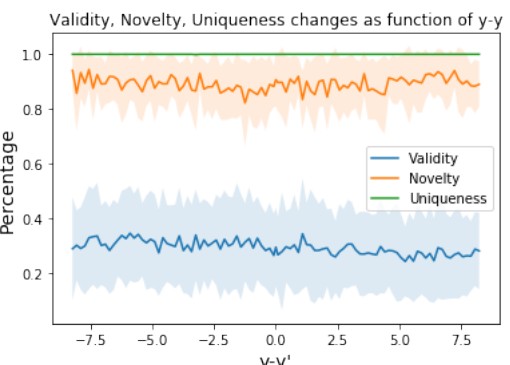

Figure 9: CGD-VAE-3-GRU model validity, novelty, uniqueness performance on property transfer task as a function of $\mathbf{y} - \mathbf{y}'$ on QM9 dataset.

Figure 10: CGD-VAE-3-GRU model validity, novelty, uniqueness performance on property transfer task as a function of $\mathbf{y} - \mathbf{y}'$ on ZINC dataset.

## 6.6 SAMPLING FROM APPROXIMATED MARGINAL POSTERIOR FOR GENERATION

With our model, during the generation, we observe that sampling from the approximated marginal posterior improves generation performance when compered to sampling from the prior. Here we

investigate if this findings holds for other baseline models or not. We explored the behavior of baselines when the **z** is sampled from the approximate marginal posterior, we observe that for CVAE and GVAE the validity did not change (as the $\hat{q}(\mathbf{z})$ and $p(\mathbf{z})$ are essentially identical); for the SD-VAE the validity increases (Table 5).

| SDVAE | | | |
|---|---|---|---|
| | Valid% | Unique % | Novel % |
| $\mathbf{z} \sim \hat{q}_\sigma(\mathbf{z})$ | 56.05 | 80.69 | 100 |
| $\mathbf{z} \sim p(\mathbf{z})$ | 43.50 | 82.26 | 100 |

Table 5: Baseline model performance on ZINC

### 6.7    MOLECULE PROPERTY OPTIMIZATION

Figure 11 is the visualization of the generated molecules from property optimization task, given in figure 6 section 4. The molecules are generated by increasing the logP of a given molecule, i.e. we hold **z** fixed and increase the **y** value. Among the all generated molecules, 19 of them are unique.

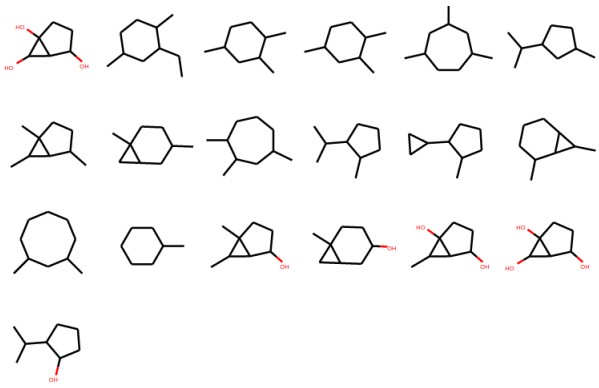

Figure 11: Molecules generated with an increasing logP

