# OpenReview forum: "Conditional generation of molecules from disentangled representations"
_ICLR.cc/2020/Conference — Reject_

### Official Review · AnonReviewer3 · 2019-10-23
**Official Blind Review #3**

**Rating:** 3

**Review:**

The authors introduce a variational autoencoder for conditional generation of molecules. The model is borrowed from text-based style transfer, applied here on sequence (SMILES) representation of molecules rather than viewing molecules as graphs (as more recent approaches). From a modeling point of view, the main new part is an additional regularizer whose role is to 1) ensure that the property used as input during generation matches the property derived from the generated molecule, and 2) to dissociate the latent molecule representation in the autoencoder (loosely speaking, its overall structure) from the property being controlled. This regularizer is just a squared difference between predicted and actual properties, averaged over independent samples from latent states and properties which are parametrically mapped to predicted properties via the decoder state (so as to be able to backprop).

The resulting ELBO criterion for the VAE, together with the regularizer, could be in principle directly used to train all the model parameters (encoder, decoder, property mapping) but apparently this criterion does not result in a "stable" estimation procedure. The model is then amended in various ways.

The first change consists of introducing a property prediction part directly into ELBO so that the posterior over latent states will also be affected by property prediction (previously property prediction would only appear in the additional regularizer). Beyond stability that the authors cite as the reason, this is likely also needed because in eq (2) the posterior over latent states given the molecule does not depend on the property. The posterior is derived from a v-structure where latent states and properties appear as independent parents of the molecule. Since for the training cases the property is fixed, this would be fine with an arbitrarily complex posterior encoding. However, when the encoder is defined parametrically, this dependence may be needed, and may relate to the stability issues. It seems a bit roundabout way of putting the dependence back in, if so. The authors should comment on this further.

The other simplification that is introduced is that the prior over the latent states in the regularizer is modified to be the average posterior of training encodings, and later further collapsed (for computational reasons) to an adaptively estimated simple Gaussian with variance that matches the average posterior variance.

It's a bit unsatisfying that the authors only use and experiment with a simple logP as the property to control in the model. This is not really a particularly relevant property to control. The approach should in principle generalize to other properties but the many adjustments needed to make it work makes this a little questionable. Also, the results are a bit limited (only logP) and not encouraging (see comments below).

- how is validity defined? SMILES syntax or validity as a molecule?

- is table 1 reconstruction really autoencoder reproduction percentage? Is the correct property for the molecule being reconstructed offered as an additional input to the authors' model? isn't this a bit unfair?

- the correlation between property given as input and the property derived from the generated molecule does not seem like a good metric. Wouldn't a model that overfits (essentially spitting out the nearest property neighbor from the training set) do particularly well according to this metric?

- it seems Figure 6 can be interpreted as also indicating some degree of overfitting?

- the stacked LSTM molecular generator baseline that takes the property as input seems to do much better in terms of maintaining the property in its generated molecules and it also realizes a more diverse collection of molecules. It's true that it doesn't support incremental molecular manipulation directly, eg, if one is keen on keeping the modified molecule close to a starting point. But it is so much better at maintaining the input property in its diverse realizations that perhaps one should instead invest in a tailored sampling procedure to obtain realizations close to a chosen reference.

The literature references seem to omit all molecular generation work since 2018


**Experience Assessment:**

I have published in this field for several years.

**Review Assessment: Checking Correctness Of Derivations And Theory:**

I carefully checked the derivations and theory.

**Review Assessment: Checking Correctness Of Experiments:**

I carefully checked the experiments.

**Review Assessment: Thoroughness In Paper Reading:**

I read the paper thoroughly.

---

> ### Author Response · Authors · 2019-11-15
> **Response to Reviewer#3**
>
> We would like to thank you for the detailed comments. We addressed each question as follows:
>
> # The dependency of the latent states on the property predictor (“The first change consists of introducing a property prediction part directly into ELBO…”)
> The ELBO in equation 6 is lower bound for p(x|y), while the extended ELBO introduced in equation 10 is lower bound for p(x, f(x) |y). This extension models not only generating molecules x for a given y, but also the corresponding property f(x) of the generated molecule — this does have the effect of explicitly encouraging the generated property f(x) to match the value of y passed to the generative model. Regarding the structure of the posterior itself, it is true in eq (10) we write in terms of a posterior purely over z, as q(z | x), with y as an additional observed latent variable in the generative model. However, one could also view the “joint” posterior q(z,y | x) as defined via q(z|x) and a delta function on the “true” property function f evaluated on the molecule x. Adding y as part of the posterior over z, i.e. with a functional form of q(z | x, y), does not really provide much additional context since y is assumed to be a deterministic property of x and the encoder network is quite powerful. In practice we saw no discernible difference between models trained with encoders q(z | x) and q(z | x, y) and thus went with the simpler option; of course, it is possible if more or different properties y are chosen this may change.
>
> # The prior and averaged posterior over the training encodings
> We would like to try to clear up some possible confusion here. We believe there is some misunderstanding between section 3.4 (describing the evaluation of the gradient) with the experiment section (describing how we sample z when doing generation at test time). Throughout training, as stated in section 3.4, instead of sampling z from regions of the prior that may correspond to invalid molecules, we instead sample from the aggregate posterior; this is motivated by the style transfer definition, and handled in practice by sampling z from the approximate posterior q(z|x_j) for x_j not paired with the target y_i given to the decoder.  During testing, for generation, we found that sampling z from the approximated aggregated posterior where it is estimated using simple Gaussian with a variance that matches the average posterior variance has better performance in terms of validity. The restriction to the simple Gaussian is made not for computational reasons (it is simple to instead refer to the original dataset to sample from the aggregate posterior) but rather for a fairer comparison with sampling from the prior.
>
> # Limited to LogP
>  Our method by design is not restricted to specific properties; adapting to other properties require no further adjustment. We wanted to make a “real” conditional generation model for molecules such that for a certain target value we can sample molecules that has properties match. Many existing models that can do “conditional” generation are actually targeted to optimize certain properties, which means, the model can only generate molecules with the target property raised. Those models make sense for, say, QED as a higher QED score is clearly better. However, for generating molecules with certain values, LogP is a more sensible choice of property, as particular values may be desired, rather than simply “as large / small as possible”. The model itself is not conceptually restricted to certain properties; all we need to do is to replace the LogP with other properties at train time.
>
> # Validity measure
> Validity is defined as an actually valid molecule, confirmed using RDkit (not just proper syntax).
>
> # Is table 1 reconstruction really autoencoder reproduction percentage? Is the correct property for the molecule being reconstructed offered as an additional input to the authors' model? isn't this a bit unfair?
> The base VAE model is in a supervised VAE, by default it takes x and y as input and try to reconstruct x — so, in this case, they both have the same information (both molecule embedding z and property y) at generation time. This highlights that the reconstruction performance of our model does not degrade, despite more successful incorporation of the property into the generative model.
> However, even for unsupervised standard VAE, it takes as input x, and tries to reconstruct x. In both model information about x is given to reconstruct x. The only difference is the first model tries to encode all information about x into z and the second model tries to encode in z all information about x beside the y.

---

> > ### Author Response · Authors · 2019-11-15
> > **Response to Reviewer#3 (continuation)**
> >
> >
> > # Figure 6 and overfitting
> > The step-like nature of this figure is actually primarily due to the fact that not every “structure” is compatible with different target values of the property. It is true that for a z that is encoded from certain x, when we combine it with different ys, for many values of y, it tends to reconstruct back the original molecule that is being encoded ( first block in figure 6). However, this is not the same as overfitting. First, note that the x that is used to encode is taken from the held-out test set. Second, the model also generated many other molecules that are different from the original x ( the other blocks in figure 6 ). The only thing in the graph is that when 100 different ys are paired with a fixed z,  the model only generates 19 unique molecules (among around 90 valid molecules). It shows that for a certain combination of z and x, physically there is  no molecules that that matches such combination or our model did not learn yet to generate molecules for such combination. This is due to the fact that molecule space is discrete, and some combinations of “style” with particular target properties may not be possible. It also shows that the effect of z is stronger than the effect of y in the generation phase: we can see that for a given z, with a very small change of y, one often end up generating the same molecules.  For example for z that is encoded from a molecule whose property is around -0.9 (after scaling to -1 and 1), if we change the property towards 0, we either get invalid molecules or the same molecule; only once we change the property to around 0.1 do we get different molecules.
> >
> > # Correlation as a measure when the model overfit
> > Before proceeding to the conditional generation, we also checked the novelty of the generated molecules when compared against the training set. As table 1 shows the novelty is pretty high (100% for ZINC and 98 % for QM9 ). Therefore, when we sample a property value from the testset and generate molecules from that particular properties, most of the generated molecules are novel.
> >
> > #The literature references seem to omit all molecular generation work since 2018
> > We did, in fact, include quite a few papers from 2018 in the literature review. Due to the space limit, we took the most relevant ones, but we can add a more recent literature review as part of an additional discussion on using our conditional generation alongside other decoder architectures.
> >
> > #Stacked LSTM works better
> > It is also to our surprise that a simple conditional LSTM on SMILES works so well for conditional generation. Perhaps even more surprising is that such a strong and simple model has not seen more widespread use as a baseline in work on conditional generation (“Learning Deep Generative Models of Graphs”, Li et al 2018, is the only example we could find). However, as it does not produce a latent representation of the molecule the way a VAE model does, direct style transfer is not possible. The setting we focus on has the potential of both doing direct conditional generation and style transfer, and it also produces continuous latent representations of the molecules which could be useful for other downstream tasks. We certainly agree that investigating ways for directly enabling style transfer on such conditional LSTMs is an interesting direction for future work.

---

### Official Review · AnonReviewer2 · 2019-10-23
**Official Blind Review #2**

**Rating:** 1

**Review:**

This paper proposes a VAE-based conditional molecular graph generation model. For that purpose, the disentanglement approach is adopted in this paper: learn to separate property information from the structure representation of a molecular graph. The authors use the supervised VAE objective since the KL regularizer in the objective has reportedly disentanglement-promoting effect. The final objective function is a standard VAE ELBO plus a penalty term of the property value prediction error. Non-differentiable property estimation is conducted via stochastic sampling expectation with the help of an external program (RDKit).

The proposed model directly optimizes the generation model conditioned on the property values in a single objective function. This is preferable compared to many existing molecular graph generations, where property optimization is often carried out after the core generative models are trained and fixed.
Derivation of a stable lowerbound (Eq. 11) is another plus.

My main concern about the paper is a weak survey for disentanglement researches. Since the core of graph generation model is not original of the authors (adopted from Dai+, 2018), the main technical advancement of the paper should be related to the disentangling VAE modeling.
Current researches in disentangling VAEs go beyond the InfoGAN and beta-VAE. I present a part of the must-referred papers below:

[Gao19] Gao+, “Auto-Encoding Total Correlation Explanation”, AISTATS, 2019.
[Kim_Mnih18] Kim and Mnih, “Disentangling by Factorising”, ICML, 2018.
[Ryu19] Ryu+, “Wyner VAE: Joint and Conditional Generation with Succinct Common Representation Learning”, arxiv:1905.10945, 2019.

Among them, [Ryu19] is closely related to the proposed framework. In my understanding, the variable dependency structure studied in this paper (Fig. 1) is studied by [Ryu19]. The authors should clearly state the novelty of the proposed work in the literature of these disentanglement VAE works.

I think the L_disent (Eq.5) is not a penalty for disentanglement: it enforces the model to correctly predict target property values, and do not say anything for factor disentanglement, correct?

I have a few concerns about the experiment designs.
In the table 1, reconstruction performance evaluations, the authors chose string(SMILE)-based molecular graph generation models. However, it is largely admitted in the molecular graph generation studies that the graph-based generative models generally performed better. I want justifications for the choice of string-based generation models. Extending the table 1 with graph?base SotA generation models will strengthen the manuscript greatly.
For example:

[Jin18] Jin+, “Junction Tree Variational Autoencoder for Molecular Graph Generation”, ICML, 2018.
[You18] You+, “Graph Convolutional Policy Network for Goal-Directed Molecular Graph Generation”, NeurIPS, 2018.

Especially, [You18] directly optimize the property values of the generated graph. This is closely related to the goal of this paper, thus a proper reference and discussions are strongly expected.


Figure 5. is an attractive visualization of the latent (z, y) vectors. However, the authors do not provide how to understand the figure. I GUESS that the authors want to show that learned z and y are somehow disentangled: the structural changes of molecular graphs are less sensitive to the vertical axis (property value) than the horizontal column (different). Please clearly state key messages of each figure and table.

There are several presentation issues. They are details, but fairly degrades the readability of the manuscript.
- Too small letters (alphabets) in Figure 3, 4, 6, it is simply unreadable so I cannot tell main messages of these figures (So I do not evaluate these figures positively).
- In table 1, what are the significance figures of the presented scores? Some scores have 3 digits, others have 4 digits. Some are rounded at 0.01 precision but others are round at 0.1 precision.

Evaluation summaries
+ A graph generation model combining conditional property optimization in a single objective function.
+ A new lowerbound for stable training (Eq.11)
+ The property and the structure of the graph are somehow disentangled in the experiment (Fig. 5).
+- Disentanglement effect is not modeled directly in the proposal. L_disent is not for disentanglement but for property regression.
-- Survey for disentangling VAEs is not enough. This makes the proposal less convincing in terms of the novelty and the contribution compared to the recent dissent-VAEs.
- Not compared with SotA graph-based molecular graph generation models.
- Key messages of figures and tables are not clearly stated (e.g. Fig.5).
-- Some figures are simply unreadable. The significance figure of the table 1 is unclear. These are formatting details but essential for readability.

**Experience Assessment:**

I have published one or two papers in this area.

**Review Assessment: Checking Correctness Of Derivations And Theory:**

I assessed the sensibility of the derivations and theory.

**Review Assessment: Checking Correctness Of Experiments:**

I assessed the sensibility of the experiments.

**Review Assessment: Thoroughness In Paper Reading:**

I read the paper at least twice and used my best judgement in assessing the paper.

---

> ### Author Response · Authors · 2019-11-15
> **Response to Review #2**
>
> We would like to thank you for the detailed comments. We addressed each question as follows:
>
> #Relation to work on disentanglement
> We want to emphasize that there is a very important difference between what we do and the disentanglement work that the reviewer mentions. These works are all doing unsupervised disentanglement, i.e. they seek to disentangle the components of the learned z latent variable. Instead what we do is disentangling the latent variable z from the observed property y, as explained above. We want to make sure that z contains no information about y. These are two very different settings, a fact that guided our decision not to make a more extensive literature review on the unsupervised disentanglement work. We can add discussion which contrasts this paper (and other semi-supervised disentanglement work) to the references suggested here.
> By disentangling z and y we are able to do directly conditional generation and style transfer, making our work the first that can directly address conditional generation tasks. Previous work that seeks to generate molecules with certain properties require additional optimization and search steps, or can only generate molecules were a target property is maximized.
>
> The work of [Ryu19] can do style transfer as well, as they present in Fig 2 in their paper. However, the problem setting is totally different from our Their work aims to learn a common representation z as a shared concept given two correlated data variables (x,y), i.e., a pair of images, and a local representation to capture the remaining randomness. As a result, it allows them to do conditional generation and style transfer, as the shared components are isolated into a single latent variable. However, the model consider a pair of correlated data and aim to learn a common representation. This would be useful if we were aiming to extract commonality from, say, a dataset of pairs of similar molecules; however, this is not applicable to our setting, where the property y is not “data” which we could posit is generated in parallel to other data x, but rather an essential property which would be an input into the generative process for x. In this case there is no “shared” representation between x and y beyond the value of y itself! We assume that the property y is a part of the main factor that generates the input data x and try to learn a representation z that captures the rest of the factor other than the properties y.
>
> #On the final objective (eq 11) and the non-differentiable property objective
> The final objective in Equation 11 does not contain a standard VAE ELBO. Its first term already derived in equation 9 does both reconstruction and property prediction jointly. The second term does the disentangling by independently sampling z for any given y_i (eq 5, see also discussion above). Moreover, we never use RDkit in our optimization and learning procedure. We operate over the decoder output, the logits, and train the property predictor f_\omega over that decoder output to map to the correct property y. This allow us to circumvent the non-differentiability issues that the discrete molecule structure and RDkit raise.
>
> #Experiments and relation to graph-based approaches
> The primary contribution of our paper is the adaptation of an existing deep generative model for molecules from an “unconditional” setting with an unstructured latent space, to a model which has structured latent variables that disentangle target properties from the remainder of the representation.
> We chose a particular SMILES-based model for these experiments, but in fact the entire process is modular. The encoder model can trivially be replaced with any other choice of encoder from other graph-based VAE-style models for molecules. The options for the decoder are only slightly less flexible, and only require a structure where the decoder is composed of real-valued neural network layers (analogous to the function g() in the paper) which are then fed into a discrete sampling process. This structure is used by most if not all recently proposed deep generative models for molecular graphs and SMILES strings which we are aware of.
> Our model can do direct conditional generation of molecules that exhibit a desired property value; neither  [Jin18] nor  [You18] can do direct property optimization. More specifically, [Jin18] every time they want to obtain a molecule with a specific property value they have to go through an additional optimization/search step, e.g. Bayesian optimization, in the latent space. [You18] follow an RL approach to molecule generation, which by construction is more difficult to train. Moreover, their reward considers whether the produced molecules match a desired target, such as a desired property value range. Every time then, that one wants to generate molecules with a different property value range, one has to retrain the model from scratch.

---

> > ### Author Response · Authors · 2019-11-15
> > **Response to Review #2(continuation)**
> >
> >
> > #On the disentanglement behavior of our model
> > We believe that there is a confusion in the understanding of our model and objective function. Let us consider the following three generative models (similar to the discussion with reviewer 1)
> > Standard VAE: 	 	x -> z -> \hat{x}
> > Supervised VAE:	 	x -> z, (z,y) -> \hat{x}
> > Our model (eq 3):		x->z (z,y) -> \hat{x}, f(\hat{x}) = y
> >
> > First, a comment on the disentangling properties of Standard VAE and Supervised VAE. Indeed the KL divergence of the standard ELBO (eq. 2) has a disentangling-promoting effect, due to the fact that it pushes the learned posterior to be close to an isotropic Gaussian. Nevertheless, what is disentangled in the standard ELBO (eq 2) is only the dimensions of the latent variable z. Instead what we want to achieve is the disentanglement of the learned latent variable z from the property y. So let us take a look on how the three models listed above operate, repeating a part of our answer to reviewer 1.
> >
> > If the property y is a function of x, and we have a perfect generative model (and corresponding encoder), i.e. one that gives us a perfect reconstruction, then the reconstructed molecule \hat{x} will match x and thus will have the property y. This means that one can just use 1) standard VAE and the reconstructed x will have the appropriate property. It also means that in 2) supervised VAE there is no reason necessarily for the decoder to consider the information brought by y, since everything is inside z. Of course we will not be able to do conditional generation and control y at will, since in the case 1) standard VAE we have no y, while in the case of 2) supervised VAE the y is potentially ignored by the decoder.
> >
> > We can speculate that this trivial behavior for 2) Supervised VAE is indeed what happens. If we look at the results we give in Table 1, where SD VAE is what we call above the standard VAE, and Sup-VAE-X-GRU is the supervised VAE, we see that these two models have almost an identical reconstruction performance which provides some evidence towards the fact that the Supervised VAE does ignore the label information y.
> >
> > In our model we add the constraint that the reconstruction \hat{x} should have the y property, i.e. f(\hat{x}) = y. However if we only train with z,y pairs, where z is an encoding of a true x and y is the property of x, i.e. what reviewer 1 calls keeping the y fixed during training, then the model can still collapse in the trivial case discussed above in which it only considers z; since z can contain all the information needed to reconstruct x and then trivially match the constraint f(\hat{x}) = y. Thus any model that has a perfect reconstruction will trivially satisfy the requirement that \hat{x} has the property y, including our model and the supervised VAE.
> > By arbitrarily switching y (last equation section 3.4)  we make sure that the decoder cannot have access to the label information through z, since z and y are now by construction conditionally independent given x. As a result, we force the decoder to rely on y. This switch originates in our disentanglement regulariser (eq 5) were one should note that y_i and z are independent. y_i comes from the training data but z is randomly sampled from its p(z) prior. Thus there is no relation between y_i and z and no way for the decoder to get the correct y from z since these are now independent by the very definition of our disentanglement regulariser (eq 5)
> >
> > #Figure 5 and figure readability
> > Regarding figure 5: To produce the figure we randomly sampled 9 QM9 molecules, the ones surrounded by the dotted rectangles. For each one of them we retrieve its latent representation z and combine it with 11 different property values and then generate the respective new molecules. For each original molecule we give the generated molecules ordered along the y axis according to the y property that they actually exhibit. The x-axis does not provide an ordering of the original molecules according to z, in fact we have ordered the original molecules by their y property. The graph should be seen column-wise. We will clarify this. And we will correct the readability of the other figures.  We updated the figure and tables in the new version to make it more readable.

---

### Official Review · AnonReviewer1 · 2019-10-23
**Official Blind Review #1**

**Rating:** 6

**Review:**

# Summary
The paper considers the problem of generating molecules with desired properties using a variant of supervised variational auto-encoders. The key novelty is the idea to learn a representation with disentangled molecular properties which can be modified during generation of novel molecules.

To account for the diversity of molecules in a particular target class the conditional probability of a molecule x given a target property y is modelled with a latent variable z, i.e., p(x | y) = \int p(x | z, y) p(z) dz.

The "style transfer" is achieved by taking an initial molecule x and computing the posterior of novel molecule x' with modified property y' via p(x' | y', x) = \int p(x' | y', z) p(z | x) dz.

The latent representation is learned using a supervised auto-encoder (Kingma et al., 2014), formally specified in Eq. (2). The optimization problem is then further extended by imposing a soft constraint that the molecules from a particular class exhibit a certain property (Eq. 3). The reason for this constraint is to enforce that the conditional probability p(x | y, z) takes into account the information in label y. This part might not be sufficiently well-motivated and would benefit from strengthening the argument for the property predictor and soft constraint (would an illustration be possible here?). In particular, why should it not be possible to encode (and keep fixed during training) the information present in y into the sufficient statistics of p(x | y, z)?

The soft constraint involves an oracle function (see Eq. 3), which evaluates the designed molecules. Section 3.2 deals with the approximation of the oracle via a property predictor function to side-step the fact that the oracle is not necessarily differentiable or CPU-bound. Section 3.3 and 3.4 deal with joint optimization of the supervised variational auto-encoder and gradient estimates.

The approach is evaluated on two datasets, relative to state-of-the-art baselines based on variational auto-encoders and deep learning. The main goals of the experiments are to establish that the approach can learn useful disentangled conditional distributions over molecules and to assess the extent of improvement in the data generation process as a result of using the soft constraint term. The focus of the experiment is on generating molecules with certain range of logP values. The data generating process is simulated/evaluated while controlling for the logP value range and molecular structure.
The experiments provide an objective assessment of the merits of the approach and indicate clear advantages over prior work.


# Recommendation
The paper is very well written, easy to follow, and properly structured. The work is novel and the idea to disentangle molecular properties from the target property is quite interesting. The experiments are detailed and compare to baselines based on variational auto-encoders and deep learning. The part that could be improved is the choice of the target property. In particular, it would be great to evaluate the approach on the problem of finding molecules binding to a certain protein site. Such a problem is likely to exhibit scaffold hops and activity cliffs which make drug design problems very difficult (e.g., see [6-8]).


# Related work (additional references)
- While mapping of the discrete space of molecules to a continuous latent space (in active learning approaches combined with VAE) allows for gradient computation and the use of active learning/optimization techniques, the very difficult problem of finding latent space pre-images persists (in general, the problem should be in the NP class). The latter refers to finding a molecule that maps to a fixed point in the latent space. There have, however, been some efficient approximations in special cases [1-3].
- In [4] and [5], an approach for generation of molecules with desired properties has been pursued with posterior sampler based on a conditional exponential family model. The representation is based on a tuple kernel mapping mapping (x, y) to some reproducing kernel Hilbert space (without disentanglement). The approach provides a consistent data generation process and a mean to deal with the fact that designs are not independent and identically distributed (e.g., Eq. 2 assumes that examples are IID).


# Black-box target property
- The logP property might not be a good proxy for the effectiveness in generating molecules with desired binding activities (to some protein site) because (to the best of my knowledge) it does not exhibit the "activity cliff property" characteristic to drug design. This refers to the property that a small change in molecular structure results in a completely different activity level (e.g., see [6-8] and [4]). In this sense, the "style transfer" might be beneficial for finding molecules binding to a certain protein site that are structurally similar to some available molecules with bad activity levels (for which a synthesis path might be known or easy to derive).
- An actual black-box property (realized via a docking program) that is expensive to evaluate served as a target property in [5]. That docking program (supplied with suitable binding/docking constraints) can provide a nice proxy for the actual generation of molecules.


# References
[1] G. H. Bakir, J. Weston, and B. Schoelkopf. Learning to find pre-images, NIPS 2004.
[2] C. Cortes, M. Mohri, and J. Weston. A general regression technique for learning transductions, ICML 2005.
[3] S. Giguere, A. Rolland, F. Laviolette, and M. Marchand. On the string kernel pre-image problem with applications in drug discovery, ICML 2015.
[4] D. Oglic, R. Garnett, and T. Gaertner. Active search in intensionally specified structured spaces, AAAI 2017.
[5] D. Oglic, S. Oatley, S. Macdonald, T. Mcinally, R. Garnett, J. Hirst, and T. Gaertner. Active search for computer-aided drug design, Molecular Informatics 2018.
[6] G. Schneider and U. Fechner. Computer-based de novo design of drug-like molecules, Nature Reviews Drug Discovery, 2005.
[7] P. Schneider and G. Schneider. De novo design at the edge of chaos. Journal of Medicinal Chemistry, 2016.
[8] J. Scannell, A. Blanckley, H. Boldon, and B. Warrington. Diagnosing the decline in pharmaceutical R&D efficiency. Nature Reviews Drug Discovery, 2012.

**Experience Assessment:**

I have published one or two papers in this area.

**Review Assessment: Checking Correctness Of Derivations And Theory:**

I assessed the sensibility of the derivations and theory.

**Review Assessment: Checking Correctness Of Experiments:**

I assessed the sensibility of the experiments.

**Review Assessment: Thoroughness In Paper Reading:**

I read the paper at least twice and used my best judgement in assessing the paper.

---

> ### Author Response · Authors · 2019-11-15
> **Response to Review #1**
>
> Thank you for the constructive feedback.  We addressed each question as follows:
>
> #Why keeping the y fixed will not work:
> Let us consider the three following generative models
> Standard VAE: 	 	x -> z -> \hat{x}
> Supervised VAE:	 	x -> z, (z,y) -> \hat{x}
> Our model (eq 3):		x->z (z,y) -> \hat{x}, f(\hat{x}) = y
> If the property y is a function of x, and we have a “perfect” generative model and encoder pair, i.e. one that gives us a perfect reconstruction, then the reconstructed molecule \hat{x} will match x and thus will have the property y. This means that one can just use 1) standard VAE and the reconstructed x will have the appropriate property. It also means that in 2) supervised VAE there is no reason the decoder *needs* to consider the information brought by y, since everything is inside z. Of course, we will not be able to do conditional generation and control y at will, since in the case of 1) standard VAE we have no y, while in the case of 2) supervised VAE the value of y may be mostly ignored by the decoder.
>
> We can speculate that this trivial behavior for 2) Supervised VAE is indeed what happens. If we look at the results we give in Table 1, where SD VAE is what we call above the standard VAE, and Sup-VAE-X-GRU is the supervised VAE, we see that these two models have almost an identical reconstruction performance which provides some evidence towards the fact that the Supervised VAE does ignore the label information y.
>
> Furthermore even if in our model we keep y fixed to its true value the model can still collapse in the trivial case discussed above in which it only considers z; since z can contain all the information needed to reconstruct x and then trivially match the constraint f(\hat{x}) = y. Thus any model that has a perfect reconstruction will trivially satisfy the requirement that \hat{x} has the property y, including our model and the supervised VAE.
>
> By arbitrarily switching y (last equation section 3.4)  we make sure that the decoder cannot have access to the label information through z, since z and y are now by construction conditionally independent given x. As a result, we force the decoder to rely on y. We will elaborate more in the text on why y cannot be held fixed.
>
> #”Activity cliff property”
> Indeed the “activity cliff” property is very pertinent. First, we note here that our model imposes no constraint on what y should be; thus we can train our model on problems that more drastically exhibit such behavior. However, we also would contest that LogP is sufficiently discontinuous with respect to molecule structure, and particularly to single-molecule (or single-character for SMILES) modifications, as to demonstrate a relevant proof of concept. In particular, note that a diverse array of structures is generated by our model for any given target LogP — the important aspect, as also related to the discussion above, is that the property y is only weakly informative as to the molecule structure.
>
> #Relation to “pre-image” related work
> Definitely the problem of finding pre-images is challenging! Part of the goal of this setting (and indeed, autoencoder setting in general) is that we do not have to explicitly perform a search to find the pre-image of the computation of an embedding, as done in e.g. [1]. Instead, the generative model / decoder network is trained to reconstruct the original input molecule from the feature embedding / latent space.  While this may still not a perfect inverse mapping of the latent space embedding (i.e. the encoder network), we note that (a) we measure this as the reconstruction accuracy in Table (1)  and find it to be quite high ( 99 % for QM9 and 88.6 % for ZINC) and (b) the hope is that the inaccuracy is well-captured by the stochasticity in the generative model, as we do not in general try to deterministically invert the feature mapping but instead provide a distribution over molecules. We’d be happy to add a bit of discussion along these lines.

---

### Decision · Program_Chairs · 2019-12-19

**Decision:**

Reject

**Comment:**

The paper aims to generate molecules with desired properties using a variant of supervised variational auto-encoders. Disentanglement is encouraged among the style factors.  The reviewers point out that the idea is nice, but authors avoid quantitative comparison with SotA Graph-based generative models.  Especially, the JT-VAE is acknowledged as a strong baseline widely in the community and is a VAE-based model, it is important to do these comparisons.